# Efficient LLM Scheduling by Learning to Rank

**Yichao Fu**[1]    **Siqi Zhu**[2]    **Runlong Su**[1]    **Aurick Qiao**[3]    **Ion Stoica**[4]    **Hao Zhang**[1*]
[1]UCSD    [2]Tsinghua University    [3]Snowflake    [4] UC Berkeley

## Abstract

In Large Language Model (LLM) inference, the output length of an LLM request is typically regarded as *not known a priori*. Consequently, most LLM serving systems employ a simple First-come-first-serve (FCFS) scheduling strategy, leading to Head-Of-Line (HOL) blocking and reduced throughput and service quality. In this paper, we reexamine this assumption – we show that, although predicting the exact generation length of each request is infeasible, it is possible to predict the relative ranks of output lengths in a batch of requests, using *learning to rank*. The ranking information offers valuable guidance for scheduling requests. Building on this insight, we develop a novel scheduler for LLM inference and serving that can approximate the shortest-job-first (SJF) schedule better than existing approaches. We integrate this scheduler with the state-of-the-art LLM serving system and show significant performance improvement in several important applications: 2.8x lower latency in chatbot serving and 6.5x higher throughput in synthetic data generation. Our code is available at `https://github.com/hao-ai-lab/vllm-ltr.git`.

## 1 Introduction

Large language models (LLMs) are increasingly becoming the backbone of many today's Internet services and applications that serve millions of users [1]. Due to the surge in demand, efficient scheduling for LLM serving is crucial to ensure high-quality service amidst numerous concurrent users competing for computing resources. For popular interactive applications such as chatbots, this means minimizing the latency that each user perceives while maximizing the overall system throughput to accommodate as many users as possible.

Under high load, LLM services that implement a first-come-first-serve (FCFS) scheduling strategy inevitably face significant Head-Of-Line (HOL) blocking, as many requests must wait for others to execute. Figure 1 illustrates a typical example of how a long request can block shorter ones in FCFS scheduling, leading to significant HOL blocking. In such scenarios, it is well-established that the shortest-job-first (SJF) and shortest-remaining-time-first (SRTF) scheduling algorithms minimize the average latency experienced across all requests. However, SJF/SRTF are seldom implemented in LLM services because they require requests to be ordered by their remaining generation lengths, which is traditionally assumed to be difficult or impossible to know ahead of time in existing systems [2, 3].

In this paper, we contend that, although accurately knowing the generation length of requests may be difficult, it is actually not needed. Rather, just knowing the *relative ordering* between request lengths is sufficient for SJF/SRTF scheduling. To this end, we propose to use the Kendall rank correlation coefficient (*Kendall's Tau*) [4] to measure the similarity between a predicted schedule and the SJF/SRTF schedule based on groundtruth generation lengths (i.e. oracle). We demonstrate that schedules with higher similarities (measured by Kendal's Tau) to the oracle generally translate to lower latencies in real-world performance (Figure. 2).

Based on this insight, we propose to optimize the request scheduling in LLM serving via learning to rank. We show that a small auxiliary model (e.g., OPT-125M [5]) can be trained to accurately rank

---

[*]Hao Zhang is the corresponding author

38th Conference on Neural Information Processing Systems (NeurIPS 2024).

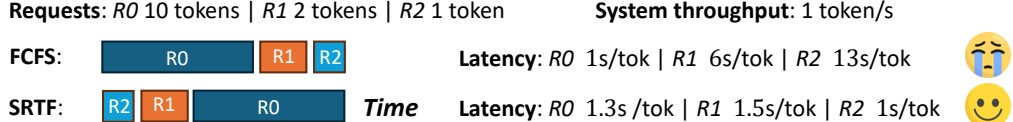

Figure 1: A long request can block short requests and introduce severe HOL blocking and high latency. We assume there is no prefill time, and the system takes 1 second to generate 1 token. With a First-come-first-serve (FCFS) schedule, the long request *R0*, which arrives first and takes 10 seconds to generate 10 tokens, will block subsequent shorter requests *R1* and *R2* for 10 seconds. Hence the latencies of *R0*, *R1*, and *R2* are $10/10 = 1, (10+2)/2 = 6, (10+2+1)/1 = 13$ s / token, respectively, perceived by users, with an average latency of $(1+6+13)/3 = 6.67$ s / token. By contrast, prioritizing shortest requests yields an average latency of $(1.3+1.5+1)/3 = 1.27$ s / token – a $5.3\times$ reduction in average latency.

LLM requests by their generation lengths, prior to execution, at virtually no cost. For both offline batch generation and online latency-sensitive tasks, by scheduling requests *on-the-fly* based on the predicted rankings, we can approximate the SRTF/SJF schedule, hence reduce average latency and improve throughput, respectively.

Compared to existing work which attempts to directly predict the generation lengths of LLM responses [6, 7], we show that our learning-to-rank approach is both more robust in approximating SRTF/SJF, hence translating to lower latency and higher throughput, but also simpler, which can be easily integrated into production serving systems (i.e., 500 LoC in vLLM).

Our contributions are summarized as follows:

- We show that knowing the relative orderings of generation lengths provides valuable guidance for optimizing the scheduling of LLM serving.

- We apply Kendall's Tau as an effective measure of the similarity between an LLM schedule and the ideal SJF/SRTF schedule, and show a higher similiary indicated by Kendall's Tau usually translates to lower latency and high throughput in practice.

- We employ *learning-to-rank* [8] to optimize the schedule and show that our method is simple and enables on-the-fly scheduling at a per-iteration basis with negligible overhead.

- Our method, when integrated with state-of-the-art serving system, significantly improves the performance on important LLM serving tasks, reducing the p90 latency of chatbot serving by $2.8\times$ and increasing the throughput of batch synthetic data generation by $6.5\times$.

## 2   Related Work

**LLM Serving Systems.**  Orca [3] introduces iteration-level scheduling and vLLM [2] applies PagedAttention, which are two key techniques for LLM serving. However, they both apply the FCFS schedule and are prone to severe HOL blocking. Scheduling for LLM serving is a relatively less explored topic. Although many LLM serving optimizations [9, 10, 11, 12, 13] have been developed recently, all these works typically assume the output length of an LLM request cannot be known before execution. FastServe [14] applies skip-join MLFQ in LLM serving. It sets up the priority of requests according to their generated length so far. Andes [15] introduces a novel quality of experience (QoE) metric for online text services, which measures human satisfaction during the whole token delivery. It employs an online preemptive scheduling method that determines which requests to execute based on scheduling objectives (e.g., average QoE) for the upcoming timeframe. Our method differs from these by predicting generation length rankings to achieve lower latency.

**Scheduling in General.** Scheduling is critical in computer systems. First-come-first-serve (FCFS) schedules requests according to their arrival time. Shortest-job-first (SJF) and its preemptive variant, shortest-remaining-time-first (SRTF), prioritize jobs with the shortest time to finish, which provably yield the lowest average latency, but may suffer from starvation problems. We discuss how to prevent starvation in §4.3. Multi-level-feedback-queue (MLFQ) maintains multiple priority queues to balance fairness and latency, but introduces substantial complexity in batch and interactive LLM workloads. Our work addresses this complexity by leveraging a simpler, prediction-based scheduling strategy.

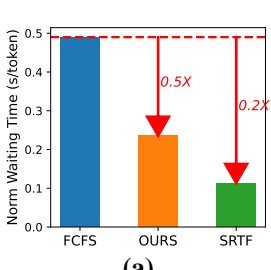
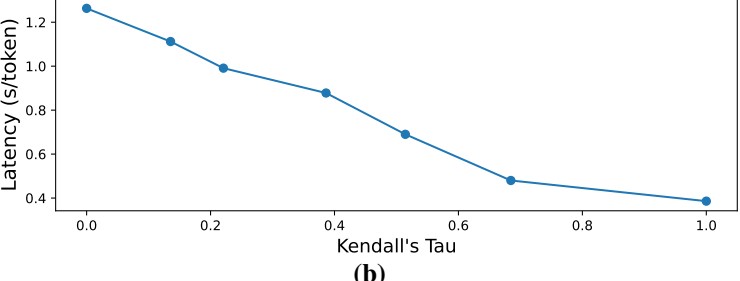

**(a)**  **(b)**

Figure 2: **(a)**: HOL blocking was evaluated by comparing FCFS and SRTF scheduling policies across 1K requests. **(b)**: Analysis revealed that higher Kendall's Tau correlation coefficients were associated with reduced latency. This finding was validated using the ShareGPT dataset with the Llama-3-8B model.

**LLM Generation Length Prediction.** Closest to our work are several recent works that predict the (exact) generation length of LLMs in order to enhance resource utilization (e.g., memory). Perception Only (PO) [7] methods let LLMs output the generation length via prompting. S3 [6], TetriServe [11] and DynamoLLM [16] use a predictor model (i.e, DistilBert [17] and OPT [5]) to predict generation length. These methods formulate the length prediction as a classification problem, whose success hinges on high predictive accuracy. Magnus [18] utilizes a language-agnostic BERT sentence embedding, a compression component, and a random forest regressor to predict generation length. Other concurrent works [19, 20] both propose a regression-based method for length prediction, fine-tuning a BERT model on the Lmsys-Chat-1M dataset with an L1 regression loss to predict the exact generation length. They tested models ranging from 300M to 3B and applied various batching policies, including no batching, dynamic batching, and continuous batching, significantly improving latency and throughput under these settings. Additionally, it supports multi-round LLM conversations. In contrast, our proposed method is built on vLLM with paged attention and uses ranking loss to optimize the predictor model. We designed a preemptive scheduling method with starvation prevention to optimize the end-to-end performance of real-world LLM serving systems.

## 3   Background

In this section, we introduce several background concepts in learning to rank, which are essential for understanding our methodology.

**Kendall Rank Correlation Coefficient.** Kendall's Tau coefficient [4], specifically the Tau-b variant we use, measures the correlation between two rankings. Its value ranges from $-1$ to $1$, where $1$ indicates perfect agreement between two rankings, $-1$ indicates complete disagreement (reversed rankings), and $0$ indicates no correlation. The formulation of Kendall's Tau is given as follows:

$$\tau = \frac{N_c - N_d}{\sqrt{(N_0 - N_1)(N_0 - N_2)}}, \qquad (1)$$

where $N_c$ and $N_d$ are the number of concordant and discordant pairs, respectively, in the two rankings. $N_0 = n(n-1)/2$, where $n$ is the total number of items. $N_1$ and $N_2$ consider tied values in each ranking: $N_1 = \sum_i t_i(t_i - 1)/2$ and $N_2 = \sum_j u_j(u_j - 1)/2$, where $t_i$ is the number of tied values in the $i^{th}$ group of ties for the first ranking and $u_j$ is the number of tied values in the $j^{th}$ group of ties for the second ranking [4]. It's important to note that tied pairs are considered neither concordant nor discordant in this calculation.

**Learning to Rank.** Learning to rank [8] is a machine learning approach applied to supervised ranking data. It is widely used in recommendation systems [21], search engine [8] and other research areas [22, 23]. Learning to rank typically takes one of three forms: pointwise, pairwise, and listwise. Pointwise turns the ranking problem into regression [24], classification [25, 26] or ordinal regression [27]. Pairwise [28, 29, 30, 31, 32, 33] method learns the relative ranking for each pair of items. Listwise [34, 35, 36, 37, 38] learns the ranking of lists of samples in a dataset.

**ListMLE.** ListMLE [37] is a listwise ranking loss used in this paper. It minimizes the likelihood function defined $\phi(g(x),y) = -\log P(y\,|\,x;g)$, where

$$P(y\,|\,x;g) = \prod_{i=1}^{n} \frac{\exp\big(g\big(x_{y(i)}\big)\big)}{\sum_{k=i}^{n}\exp\big(g\big(x_{y(k)}\big)\big)} \tag{2}$$

Here, $P(y\,|\,x;g)$ represents the probability of the permutation $y$ given the input $x$ and the scoring function $g$. $x_{y(i)}$ denotes the element in $x$ that corresponds to the $i$-th position in the permutation $y$. The idea is to maximize the likelihood of the correct ranking $y$ by using the scoring function $g$ to predict the ranking of the input $x$. The loss function $\phi(g(x),y)$ minimizes the negative log-likelihood of this probability, encouraging the model to predict a ranking close to the true ranking. ListMLE's focus on list ranking aligns with Kendall's Tau, which measures the correlation between two rankings. This ensures that minimizing the loss can help improve Kendall's Tau.

In the following section, we will introduce how we apply these learning to rank concepts to LLM scheduling, building upon the foundation laid here.

## 4   Method

### 4.1   Problem Formulation

For a given batch of requests, we define the ground truth generation length as $l$, where $l_i$ is the generation length of the $i$-th request in the batch. From this length list, we can obtain a ranking list $r$, where $r_i$ is the rank of $l_i$ within the whole batch $l$.

Our goal is to approximate true SJF/SRTF scheduling using these rankings to alleviate HOL blocking (Fig. 2 **a**) and obtain a relatively low latency in LLM serving. Different from the previous methods which target to predict the real generation length $l$, we make predictions on the ranking list $r$. The prediction of the ranking list is defined as $p$ (generated by a predictor $P$). We compute the ranking metric Kendall's Tau [4] to measure the correlation between $p$ and $r$. A Kendall's Tau of 1 means the prediction $p$ perfectly aligns with the ground truth $r$, hence we can use it to achieve perfect SJF/SRTF execution order. Conversely, A Kendall's Tau of 0 suggests no correlation between $p$ and $r$. An example is FCFS: the execution order (i.e., by arrival time) is not correlated with the generation length.

A higher Kendall's Tau reflects a more accurate rank prediction against the oracle (i.e., SJF/SRTF), which empirically translates into higher end-to-end performance, as evidenced in Fig. 2 **b**. Hence, our goal is to optimize the predictor model $P$ to generate predictions with a larger Kendall's Tau, which are more correlated to the ground truth. However, Kendall's tau is inherently non-continuous and difficult to optimize directly. To overcome this, we apply a listwise ranking loss *ListMLE* to optimize the predictor $P$. ListMLE considers the entire list of items simultaneously and treats items at all positions with equal importance, providing a more holistic evaluation of the ranking order compared to other alternatives such as pairwise and pointwise losses.

### 4.2   Generation Length Ranking Predictor

For our predictor $P$, we utilize a small OPT model as the backbone, capable of processing natural language prompts as input and generating a score for ranking. While previous methods [7, 6, 11] use classification (with bucketing) to generate accurate output length predictions, we find this approach both challenging and unnecessary. Instead, the relative ranking suffices. Based on this insight, we apply learning to rank to train the OPT model. This section explains how we train the OPT model as the predictor $P$ to rank prompts by their expected generation length.

**Predictor Structure**. The original OPT model can not directly output a score. To address this, we append a linear layer to map the hidden states of the last layer to a floating-point number, which serves as the ranking score.

**Training Data**. We aim to train the OPT model to rank prompts according to their expected generation length when processed by a target LLM (e.g., Llama-3-70B). To achieve this, we first obtain full generations by feeding prompts into the target LLM and recording the number of generated tokens. When generating model outputs, we sample tokens with a temperature of 1.0, consistent with our evaluation methodology (§5.1). The following is an example of the training data structure.

```
"prompt": "Divide 10 by 4 and remove the remainder.\n"
```

```
"output": "\nAnswer: 2 with a remainder of 0."
"output_tokens_length": 12
```

After obtaining the generation lengths, we convert them to labels representing the ranking. The simplest way would be to rank the generation lengths directly within the entire training batch and use these rankings as labels. However, recognizing that LLM generation involves some randomness due to sampling in real-world serving, we introduce a more robust approach. We bucket the generation lengths in increments of 10 tokens, then rank these processed lengths to create our training labels.

**Training**. We train the OPT on 10k samples with a batch size of 32 for 5 epochs. We employ the ListMLE loss and the Adam optimizer with a constant learning rate of $2e-5$, $\beta_1 = 0.9$, and $\beta_2 = 0.999$. To accommodate OPT's context length limitations, we truncate prompts to a maximum of 2,048 tokens.

The use of ranking loss offers several advantages. First, ranking loss focuses on correct ordering rather than precise classification, making it more robust when dealing with batches of requests where the output length distribution for each bucket is uneven. In contrast, classification loss typically relies on bucket labels for training, which can lead to poor predictive performance for minority buckets in imbalanced datasets. Second, ranking loss ensures a more reasonable bucket size, while classification loss attempts to make the predicted labels as close to the actual labels as possible. This naturally leads to the pursuit of larger bucket sizes, which is not beneficial for scheduling. Finally, ranking loss can reduce the risk of overfitting. Classification loss forces the model to minimize classification errors on the training requests, which may not generalize well to requests with covariate shifts and can cause the model to be highly sensitive to bucket size (see our study in Tab. 3).

### 4.3 Request Scheduling with Rankings

We propose a simple but effective algorithm, for scheduling requests using ranking information, as detailed in Algorithm 1. The core idea is to iteratively run the predictor model $P$ to score new requests, then sort all requests according to their predicted generation length rankings. We form a running batch based on this sorted order, subject to memory or batch size constraints. To prevent the starvation of long requests, we've incorporated additional mechanisms, which we'll explain shortly. This ranking-based scheduling algorithm operates at the iteration level, making it compatible with established LLM serving techniques such as continuous batching [3] and PagedAttention [2].

**Starvation Prevention.** While SJF/SRTF scheduling can improve overall latency, it may lead to starvation for long requests, causing users to wait excessively for responses. Different from previous fairness-promoting design [39], which focuses on the fairness between different clients, we propose a *max_waiting_time* fairness metric to evaluate the fairness at per-request level (hence reflecting per-user satisfaction). We define *max_waiting_time* fairness by considering both *Time To First Token* (TTFT) and *Time Per Output Token* (TPOT) [12] in LLM serving as follows:

$$max\_waiting\_time = \max(TTFT, \max(TPOT)). \tag{3}$$

Intuitively, *max_waiting_time* characterizes the maximum time interval a user experiences between receiving two tokens after sending a request to the server. A larger *max_waiting_time* indicates a longer waiting time for the user to obtain a response, signifying more severe starvation.

To mitigate starvation, our algorithm implements the following mechanism: 1) For each scheduling step, we increment a request's starvation count ($StarvationCount$) if it is not executed. 2) When a request's starvation count reaches a pre-defined threshold ($StarvationThreshold$), we will promote this request's priority by allocating "quantum" of execution time. 3) The request maintains this elevated priority until it exhausts its allocated quantum ($PriorityQuantum$). This simple yet effective method prevents starvation at the request level, improves *max_waiting_time*, and ensures user satisfaction, as demonstrated in our experiments (§5.5).

## 5 Evaluation

In this section, we evaluate our proposed method against several baselines and assess the effectiveness of each component. Our results demonstrate that our method achieves state-of-the-art performance in terms of both Kendall's Tau and end-to-end serving performance metrics: latency and throughput. Notably, we achieved a $2.8\times$ lower latency in chatbot serving and a $6.5\times$ higher throughput in synthetic data generation.

**Algorithm 1** Ranking Scheduler

---

1: **Input:** request queue $Q$, predictor model $P$, LLM $M$, hyper-parameter $StarvationThreshold$ prevents request's starvation, hyper-parameter $PriorityQuantum$ limits request's priority time
2: **while** True **do**
3:     Receive batch of new requests $N$
4:     **for** $r$ **in** $N$ **do**
5:         $r.Score = P(r)$ {Batch Run Predictor}
6:     **end for**
7:     Append $N$ request into $Q$ upon arrival
8:     $S = Sort(Q)$ according to the pair $(r.Priority, r.Score)$ {User-defined sort function}
9:     $B \leftarrow \emptyset$ {$B$ is the running batch of the current step}
10:     **for** $r$ **in** $S$ **do**
11:         **if** $B$ is not full **then**
12:             $B \leftarrow B.append(r)$
13:             $r.StarvationCount = 0$ {Reset $StarvationCount$ when scheduled}
14:             **if** $r.Priority$ **then**
15:                 $r.Quantum = r.Quantum - 1$
16:             **end if**
17:         **else**
18:             $r.StarvationCount = r.StarvationCount + 1$
19:         **end if**
20:     **end for**
21:     **for** $r$ **in** $Q$ **do**
22:         **if** $r.StarvationCount \geq StarvationThreshold$ **then**
23:             Promote$(r.Priority)$ {Promote $r$'s priority and assign quantum}
24:             $r.StarvationCount = 0$
25:             $r.Quantum = PriorityQuantum$
26:         **else if** $r.Priority$ and $r.Quantum \leq 0$ **then**
27:             Demote$(r.Priority)$
28:         **end if**
29:     **end for**
30:     Execute $B$ with $M$
31:     Remove finished requests from $Q$ and output
32: **end while**

---

## 5.1 Evaluation Setup

**Testbed.** Our end-to-end evaluation testbed consists of a DGX server with 8 NVIDIA A100 40GB GPUs, 256 vCPUs, and 1TB host memory. The GPUs are interconnected via NVLink.

**Serving Models.** We utilize the latest Meta Llama-3 models in two sizes: 8B and 70B [40]. All experiments use FP16/BF16 precision, which is the most common setting in LLM deployment. The 8B model runs on a single GPU, while the 70B model runs on 8 GPUs with tensor parallelism [41].

**Workloads.** We evaluate using the ShareGPT [42] and LMSYS-Chat-1M [43] datasets, which comprise open-ended, real-world conversations with proprietary LLM chatbots such as ChatGPT [1] and Claude, as well as 25 other open-source LLMs. For each dataset and model pair, we sample 10k non-overlapping prompts for serving and another 10k for training the ranking predictor. The length distributions of the datasets are provided in Appendix B. Model generations are conducted using random sampling with a temperature of 1.0, ensuring consistency during predictor training and serving evaluation. It's worth noting that our framework is insensitive to the sampling parameters.

**Evaluation metrics.** For chatbot serving, we measure average and p90 per-token latency, which is the per-request latency divided by the output length. For offline synthetic generation tasks, we use throughput (requests/second) to indicate request generation speed.

**Scheduler Settings.** We compare our method (i.e., **ranking predictor**) with four baselines implemented on top of vLLM v0.4.1:

- **FCFS**: A First-Come-First-Served scheduler that supports executing prefill and decode in the same step. For each scheduling step, it selects requests by earliest arrival time.

- **MLFQ**: We implement a Multi-Level Feedback Queue in 1.2k lines of Python code on vLLM. This scheduler leverages chunked prefill from vLLM to run prefill and decode in the same step, as described in FastServe [14]. The implementation's correctness is validated in Appendix A.

- **Perception Only (PO)**: We implement Perception Only [44] on vLLM, enabling the LLM to self-predict its token generation length. The implementation consists of two phases: First, we configure the LLM to generate 15 tokens (half of the maximum token count used in [44]) following a FCFS policy, using this output to determine the predicted generation length. Second, after obtaining these predictions, we schedule subsequent requests based solely on the predicted lengths.

- **Classification**: We train a classifier using an OPT model as a backbone. For Llama-3-8B, we use the OPT-125m model, and for Llama-3-70B, we use OPT-350m, which can be supported by 8-way tensor parallelism. Following the setting in S3 [6], we use 10 buckets with a bucket size of (max context length / number of buckets) for high classification accuracy. We map the hidden states of the OPT model to the number of buckets with a linear layer and use the same training settings as in §4.2 but with a cross-entropy loss.

- **Ranking (Ours)**: We implement our ranking scheduler (described in §4.3) with the ranking predictor and training configuration detailed in §4.2. The implementation uses an OPT model of identical size to that used in the classification method.

## 5.2 Chatbot Serving Scheduling

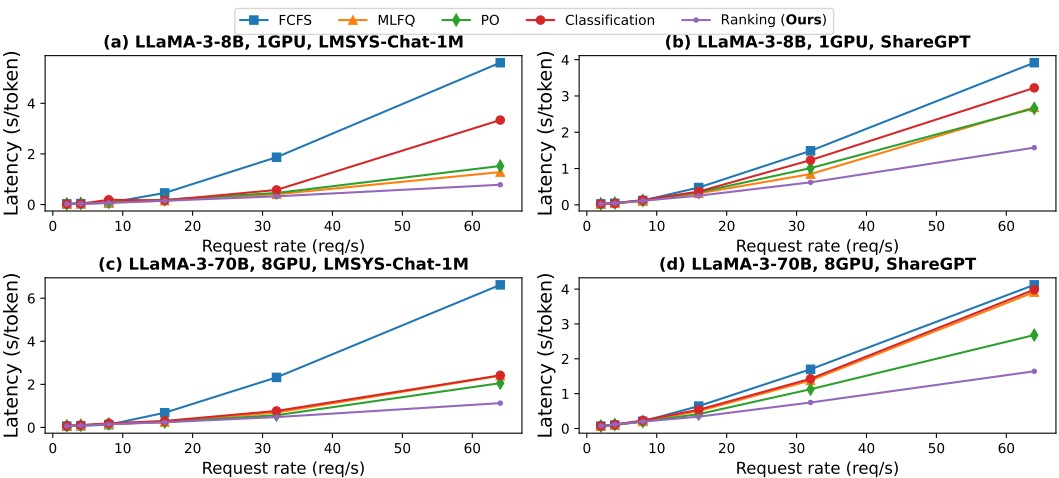

Figure 3: Mean latency of different schedulers with Llama-3 models on real workloads.

Fig. 3 compares the latency of our proposed ranking method with four baseline methods on ShareGPT and LMSYS-Chat-1M datasets with increasing arrival rates [2, 14, 12]. Under a rate of 64 requests/second, our method improves mean latency by up to $6.9\times$ compared to FCFS and $1.5\times$–$1.9\times$ compared to PO. MLFQ and PO still face severe HOL blockings as they must run *all* requests for a certain time to obtain information for scheduling. PO must execute all arriving requests with the LLM to generate a length prediction. MLFQ must run all arriving requests before they enter the next priority level. The classification method optimizes for accuracy instead of ranking, missing optimization opportunities. While classification and our method still need to process all the requests first to obtain a prediction, using an OPT model takes less than 2% of the time (as shown in §5.5), thus greatly reducing HOL blocking.

**Handling buristiness**. We evaluate our method's performance under bursty workloads, where users suddenly submit many requests to the LLM server [45, 46]. Tab. 1 compares the latency of our method against baselines with a burst of 2k requests. Our proposed ranking method significantly improves latency, achieving up to $2.0\times$ lower mean latency improve and $2.8\times$ lower P90 latency compared to PO.

Table 1: Latency (s/token) with Burst of 2K requests

| Model | Dataset | Mean Latency (s/token) | | | | | P90 Latency (s/token) | | | | |
|-------|---------|------|------|------|--------|------|------|------|------|--------|------|
| | | FCFS | MLFQ | PO | Class. | **Ours** | FCFS | MLFQ | PO | Class. | **Ours** |
| Llama-3-8B | ShareGPT | 1.15 | 1.07 | 1.35 | 1.13 | **0.56** | 1.60 | 1.57 | 1.67 | 1.51 | **0.67** |
| Llama-3-8B | LMSYS-Chat-1M | 1.73 | 0.80 | 0.75 | 1.77 | **0.38** | 4.86 | 1.56 | 1.47 | 4.98 | **0.52** |
| Llama-3-70B | ShareGPT | 1.44 | 1.37 | 1.04 | 1.26 | **0.78** | 2.01 | 1.89 | 1.35 | 1.73 | **0.96** |
| Llama-3-70B | LMSYS-Chat-1M | 2.17 | 1.00 | 0.95 | 2.23 | **0.54** | 5.54 | 1.91 | 1.72 | 5.72 | **0.82** |

## 5.3 Synthetic Data Generation Scheduling

Synthetic data generation (SDG) is emerging as an important inference workload due to the data-hungry nature of LLMs. In SDG, shorter responses are often preferred for several practical reasons: First, generating concise conversations is more cost-effective given the large volume and diversity of samples required [47] in SDG. Second, longer generations can introduce evaluation metric bias [48, 49, 50]. Consequently, samples with shorter generation lengths are often preferred for model training in specific applications.

Our proposed method can improve generation throughput in these scenarios by prioritizing shorter responses. We conducted two experiments to validate this approach: 1) We established a quantity threshold (i.e., 1k requests) and measure how long the schedulers need to generate such samples given 10k prompts. 2) We set a time constraint (5 minutes) and evaluated the number of samples each scheduler could generate from the same prompt pool. The results are presented in Tab. 2. The classification method underperformed compared to FCFS due to the overhead of preprocessing 10k prompts with the OPT model and its limited ability to identify shorter requests. In contrast, our proposed method effectively prioritized shorter requests, achieving a $2.4\times$-$6.5\times$ reduction in generation time for 1k requests and up to $3.2\times$ improvement in throughput within the 5-minute window. However, it's important to note that in scenarios where shorter generations are not preferred, the throughput improvements would be minor.

Table 2: Throughput Improvement with Proposed Ranking Method

| Model | Dataset | Time (s) To Generate 1k Samples | | | Generated #samples within 5min | | |
|-------|---------|------|----------------|----------------|------|----------------|----------------|
| | | FCFS | Classification | Ranking (**Ours**) | FCFS | Classification | Ranking (**Ours**) |
| Llama-3-8B | ShareGPT | 343.29 | 421.92 | **143.18** | 841 | 655 | **1706** |
| Llama-3-8B | LMSYS-Chat-1M | 197.38 | 237.40 | **30.48** | 1348 | 1644 | **4434** |
| Llama-3-70B | ShareGPT | 440.71 | 512.84 | **231.59** | 670 | 479 | **1299** |
| Llama-3-70B | LMSYS-Chat-1M | 253.68 | 338.83 | **59.67** | 1167 | 895 | **3710** |

## 5.4 Comparing Ranking Predictors

We show that the accuracy of the targeted classification method is suboptimal for LLM scheduling. Tab. 3 compares the prediction ability of the classification method with different bucket sizes. We evaluate the classification metric (i.e., accuracy) for the classification method and the ranking metric (i.e., Kendall's Tau) for all methods on the same randomly sampled test set. A larger bucket size shows better accuracy but does not necessarily indicate a higher Kendall's Tau.

We also evaluate the end-to-end performance of these methods. The "Lat." column shows the mean latency to process 2k bursts of requests as in §5.2. The "Time" column shows the time to generate 1k synthetic data as in §5.3. A method with a higher Kendall's Tau correlates with lower latency, as proposed in §3. The time to generate 1k synthetic data is less related to Kendall's Tau, as a high Tau with a large bucket size does not necessarily mean the predictor can correctly select the shortest requests.

PO achieves higher Kendall's Tau on the LMSYS-Chat-1M dataset. However, it needs to use the LLM itself to process all requests and generate a few tokens first for prediction, which introduces a very large HOL overhead compared to light predictor-based methods, despite its good performance in terms of Kendall's Tau. In all other settings, our proposed ranking method outperforms all other methods in terms of ranking metrics and end-to-end performance.

**Generalization Ability across Distribution Shifts.** We evaluate the predictor's performance under data distribution shifts by using the LMSYS-Chat-1M dataset to test the predictor trained on ShareGPT, and vice versa. The predictor trained on ShareGPT achieves a Kendall's Tau of 0.54 on ShareGPT but

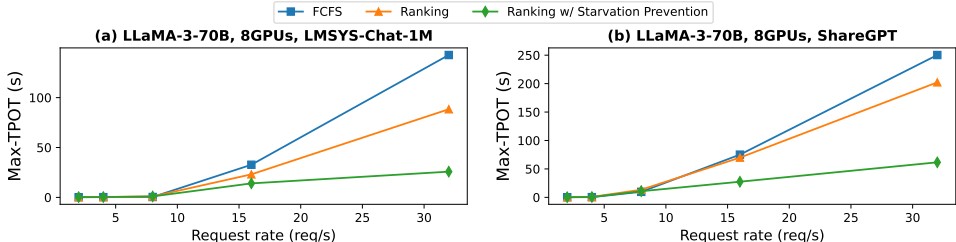

Figure 4: Average *max_waiting_time* across all requests with different scheduling method

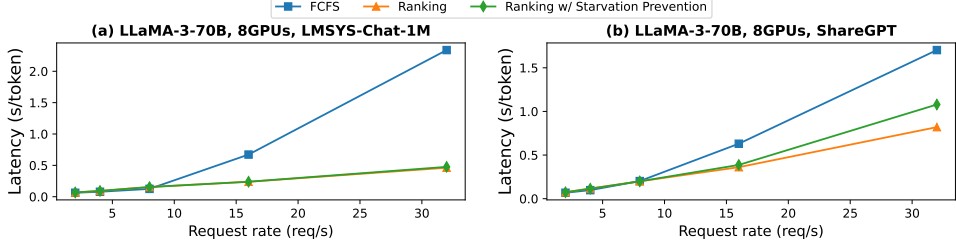

Figure 5: Influence of starvation prevention on latency

drops to 0.45 when tested on LMSYS-Chat-1M. Conversely, the predictor trained on LMSYS-Chat-1M achieves a Kendall's Tau of 0.62 on LMSYS-Chat-1M but decreases to 0.40 when tested on ShareGPT.

Although the predictor experiences performance degradation, it still retains predictive capability, demonstrating a certain level of generalization ability. In real-world scenarios, we can mitigate the impact of distribution shifts by periodically retraining the model with historical data to maintain good ranking prediction performance.

Table 3: Ranking prediction ability with different classification (Class. in table) settings (i.e., different bucket sizes) for Llama-3-70B. Lat. column shows the mean latency processing a burst of 2k requests for chatbot serving. Time column shows the time to generate 1k requests for synthetic data generation. Optimal Prediction is using the generation length of one random seed to predict the length of another seed. Note that the p-values of Kendall's Tau are below a given significance level (i.e., 1e-3) in all settings.

| | ShareGPT | | | | LMSYS-Chat-1M | | | |
|---|---|---|---|---|---|---|---|---|
| Method | Acc. (%) | Tau (↑) | Lat. (s/tok.) | Time (s) | Acc. (%) | Tau (↑) | Lat. (s/tok.) | Time (s) |
| *Optimal Prediction* | / | 0.74 | 0.46 | 102.04 | / | 0.84 | 0.34 | 34.60 |
| Ranking (**Ours**) | / | **0.54** | **0.78** | **231.59** | / | 0.62 | **0.54** | **59.67** |
| Class. (#Buckets=10) | 85.1% | 0.24 | 1.26 | 512.84 | 96.8% | 0.17 | 2.23 | 338.83 |
| Class. (Bucket Size=100) | 28.1% | 0.49 | 0.84 | 265.91 | 43.4% | 0.58 | 0.77 | 101.61 |
| Class. (Bucket Size=10) | 4.7% | 0.46 | 0.86 | 272.13 | 14.5% | 0.57 | 0.61 | 78.84 |
| Class. (Bucket Size=1) | 1.0% | 0.32 | 1.00 | 341.63 | 7.3% | 0.50 | 0.68 | 92.93 |
| PO | / | 0.51 | 1.04 | >600 | / | **0.67** | 0.95 | 322.13 |

## 5.5 Effectiveness Analysis

**Effectiveness of Starvation Prevention.** We show that our proposed starvation prevention method (§4.3) greatly reduces starvation, as measured by *max_waiting_time*. Fig. 4 shows that mean *max_waiting_time* is reduced by up to $3.4\times$ on LMSYS-Chat-1M and up to $3.3\times$ on ShareGPT compared to not using starvation prevention. Fig. 5 illustrates that starvation prevention has minimal side effects on latency, with less than 10% overhead in most cases and less than 30% in all cases, which is an acceptable trade-off.

**Overhead of Predictor Model.** Tab. 4 illustrates the overhead of the ranking predictor in responsing 1k requests. "Prefill Time" is measured by only processing the prompts with the original LLM. The overhead of the ranking models (only processing the prompts) is less than 2% in all settings. The overhead on the ShareGPT dataset is slightly higher (i.e., 1.11% and 1.69%) because the prompt length of ShareGPT is longer, as shown in Appendix B. The execution time of OPT is 10%~15% of the execution time of the original LLM in processing the prompts, largely alleviating the HOL blocking cost in length prediction compared to PO in chatbot servings.

Table 4: Overhead of Predictor Model

| Model | Dataset | Overall Time (s) | Prefill Time (s) | Predictor Time (s) | Overhead (%) |
|-------|---------|------------------|------------------|--------------------|--------------|
| Llama-3-8B | ShareGPT | 254.23 | 22.34 | 2.81 | 1.11 |
| Llama-3-8B | LMSYS-Chat-1M | 127.82 | 7.50 | 1.03 | 0.81 |
| Llama-3-70B | ShareGPT | 419.74 | 46.06 | 7.09 | 1.69 |
| Llama-3-70B | LMSYS-Chat-1M | 211.30 | 15.44 | 2.46 | 1.16 |

## 6 Limitations

**Limitation of the Ranking Metric.** Although Kendall's Tau is a widely used ranking metric, it has limitations when it comes to reflecting end-to-end performance. For example, consider a ranking prediction that accurately reflects generation length. If we randomly shuffle the predictions within the shortest 70% of requests and separately shuffle those within the longest 70%, the Kendall's Tau score will still be 0.5 for both. However, this leads to a significant latency difference, with the Llama-8B model on the ShareGPT dataset showing a $1.8\times$ increase in latency. In contrast, when the ranking is uniformly shuffled across the entire list, Kendall's Tau exhibits a stronger correlation with latency, as shown in Fig. 2.

**Limitations of the Proposed Ranking Scheduler.** The proposed ranking scheduler is designed to work with standard LLM serving techniques, such as continuous batching and paged attention. However, it has not yet been fully tested with newer optimizations like chunk-prefill [13] and prefill-decode disaggregation [12]. Future work will focus on integrating the scheduler with these advanced techniques to assess their combined performance benefits.

## 7 Conclusion

In this paper, we propose a method to train a predictor that learns to rank the generation length of LLM responses based on the given prompts using a *learning-to-rank* approach. We implement a rank-based scheduler on top of vLLM, demonstrating significant improvements across various tasks. Specifically, our method reduces latency by 2.8x in chatbot serving and increases throughput by 6.5x in synthetic data generation. Given the simplicity and effectiveness of our approach, we believe it can be easily integrated into production-level LLM serving systems, reducing serving latencies while enhancing service quality.

## Acknowledgments and Disclosure of Funding

We extend our gratitude to Junda Chen, Yinmin Zhong, and Zhuohan Li for their valuable feedback. We also thank the anonymous reviewers for their insightful and constructive comments.

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

# A Implementation of MLFQ

We are validating the correctness of MLFQ implementation by presenting the relationship of *finish time* and *output length* of requests as shown in Fig. 6. This presents a burst of 1k requests with an MLFQ base quantum of 16 seconds, the quantum growth rate of 2, and a max requests limitation of 256 for each step in the vLLM scheduler.

These rectangular blocks, whose edge lengths grow exponentially with the quantum growth rate, represent requests that are completed in queues of varying priorities. When requests from higher priority fail to fill the entire sliding window, those from lower priority begin to be processed, resulting in different blocks being adjacent to one another.

The max request limitation for each step is like a sliding window on all requests. According to the property of MLFQ, requests within the sliding window have two ways out 1) *Finish and pop out* marked by a linear increase in output lengths over time; 2) *Timeout and demote*, occurring when the finish time reaches a multiple of the quantum for the current queue, a batch of requests that arrive at the same time will be demoted simultaneously. With a short quantum for the priority queue, most requests are likely to be demoted rather than completed within the quantum, which explains the clear line trend for the first block shown in the figure. When the finish time reaches multiples of the base quantum (16 seconds in this figure), a new linear growth line appears caused by batch timeout demotions.

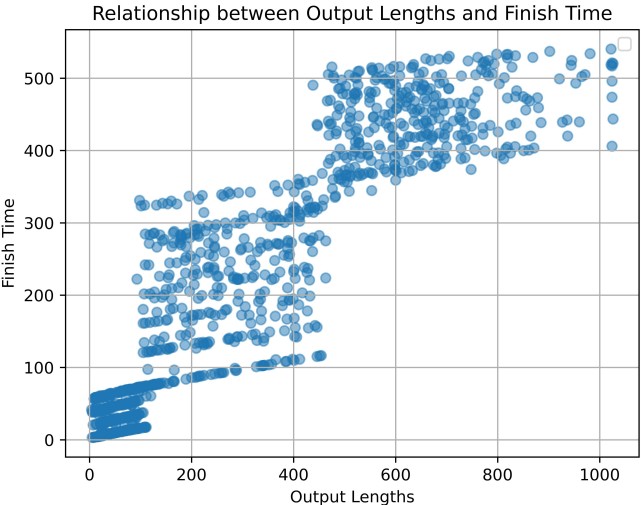

Figure 6: Finish Time of Requests with MLFQ Scheduler.

# B  Dataset Length Distribution

We randomly sample 10k samples and present the dataset distribution as in Fig. 7. We compute the input length by appending the chat template onto the prompts. We have a mean value of 85 input tokens for LMSYS-Chat-1M and a mean value of 240 input tokens for ShareGPT, which is $3\times$ longer than LMSYS-Chat-1M. The output length of the ShareGPT dataset is 100 tokens more than the LMSYS-Chat-1M dataset. On average, the 70B version of Llama-3 has a slightly longer output length (i.e., around 15 tokens).

Figure 7: Dataset Length Distribution

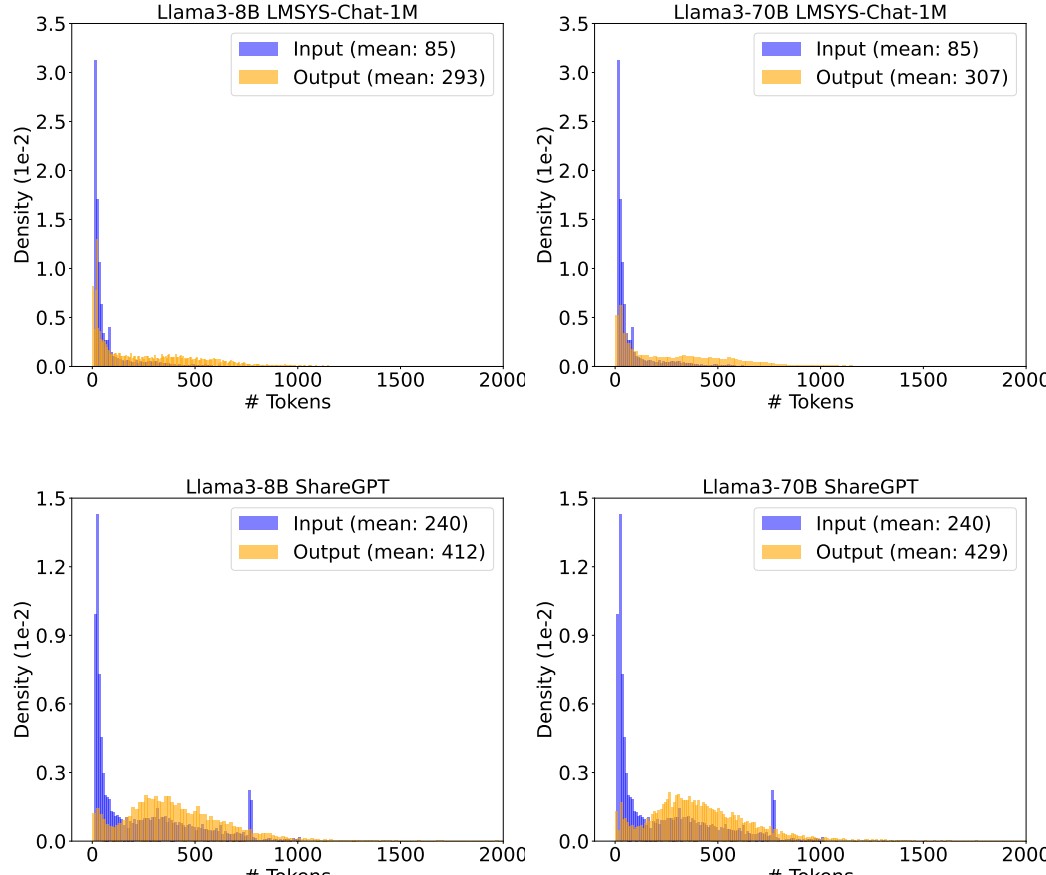

# C  Predictor's Sensitivity to Batch Size

The ranking scheduler is insensitive to batch size variations. We assess the predictor's sensitivity to batch size on the LMSYS-Chat-1M dataset, as detailed in Tab. 5. We use the predictor to calculate Kendall's Tau for various batch sizes and derive the mean and variance across the entire dataset. This experiment shows that Kendall's Tau remains within a narrow range across different batch sizes. Additionally, our method addresses severe HOL problems when there are numerous requests, in which case, the batch size is often sufficiently large for the predictor to be effective and robust.

Table 5: Predictor's Sensitivity to Batch Size

| Batch Size | Kendall's Tau Mean | Kendall's Tau Variance |
|---|---|---|
| 8 | 0.619 | 0.04 |
| 16 | 0.625 | 0.02 |
| 32 | 0.624 | 0.008 |
| 64 | 0.625 | 0.0007 |
| 128 | 0.619 | 0.001 |

## D Relationship Between the ListMLE Loss and the Kendall's Tau

The ListMLE loss defines a parameterized exponential probability distribution over all scores (as given by the model) and formulates the loss function as the negative log likelihood of the ground truth ranking $y$. Meanwhile, Kendall's Tau measures the ordinal association between the scores (as given by the model) and the ground truth ranking $y$. It is challenging to accurately describe the relationship between the likelihood and ordinal association. However, we provide an analysis demonstrating that minimizing the ListMLE loss can help improve Kendall's Tau.

To simplify the problem, we assume there are no ties between any two items, meaning each pair should be either concordant or discordant. In this case, Kendall's Tau is defined as $\tau = \frac{N_c - N_d}{n(n-1)/2}$, where $N_c$ and $N_d$ are the number of concordant and discordant pairs in two rankings, and $n$ is the total number of items. As $N_d$ increases, $N_c$ decreases because the sum of $N_c$ and $N_d$ is fixed. Consequently, we have $\Delta\tau = \frac{4\Delta N_c}{n(n-1)}$, where $\tau$ increases when $N_c$ increases.

ListMLE loss is defined as $\phi(g(x),y) = -\log P(y\,|\,x;g)$, where $P(y\,|\,x;g)$ represents the likelihood of the ground truth ranking $y$. As the likelihood of the ground truth ranking $y$ increases, the loss decreases. Although the increase of $P(y\,|\,x;g)$ does not guarantee that $N_c$ increases, the increase in the likelihood of the ground truth ranking should generally lead to a greater agreement between the ground truth ranking and the scores given by the model, which implies an increase in the number of concordant pairs (or $N_c$) and a decrease in the number of discordant pairs (or $N_d$) between the scores and the ground truth. Thus, minimizing the loss can help improve Kendall's Tau.

We further illustrate this relationship by tracking Tau and loss throughout the training process, as shown in Tab. 6. The Pearson correlation coefficient between Tau and loss is -0.9, which means that ListMLE loss and Kendal's Tau coefficient are highly negatively correlated.

Table 6: Relationship Between the ListMLE Loss and the Kendall's Tau

| Step | Kendall's Tau | Loss |
|---|---|---|
| 20 | 0.44 | 77.79 |
| 40 | 0.51 | 75.73 |
| 60 | 0.53 | 72.61 |
| 80 | 0.54 | 70.14 |
| 100 | 0.55 | 70.59 |
| 120 | 0.53 | 70.09 |
| 140 | 0.56 | 67.01 |
| 160 | 0.59 | 69.94 |
| 180 | 0.59 | 70.88 |
| 200 | 0.57 | 68.84 |
| 220 | 0.59 | 68.67 |
| 240 | 0.61 | 66.90 |
| 260 | 0.58 | 67.23 |
| 280 | 0.56 | 68.71 |

# E    The Performance Gap Between The Proposed Method and Oracle

Due to noise and randomness in the sampling process, in this section we define the Oracle as utilizing sampling results from one seed to guide the scheduling of another sampling, which represents the best performance achievable given one sampling result. The performance gap between the ranking-based method (ours) and the Oracle varies depending on the evaluation dataset. On certain datasets, our proposed method can perform as well as the Oracle. For instance, when tested on the Alpaca [51] dataset with the Llama-8B model, our proposed method closely approximates the Oracle in terms of Kendall's Tau and end-to-end latency for a burst of 2K requests, as depicted in Tab. 7. These tests were conducted on a single A100 80GB GPU.

On datasets such as LMSYS-Chat-1M and ShareGPT, there remains a small gap between the proposed ranking-based method and the Oracle. The comparison between the ranking-based method (indicated as "Ranking (Ours)") and the Oracle (indicated as "Optimal Prediction") is presented in Tab. 3.

Table 7: Relationship Between ListMLE Loss and Kendall's Tau

|        | Kendall's Tau | Latency (s/token) |
|--------|---------------|-------------------|
| Ours   | 0.73          | 0.28              |
| Oracle | 0.72          | 0.24              |
| FCFS   | 0.0           | 1.36              |

# F    Influence of The Predictor Size

Our results show that the model size has a minor effect on the prediction ability, as indicated in the following Tab. 8:

The choice to use an OPT-350m model for Llama-70B model is primarily driven by deployment considerations. The OPT-350m model, with 16 attention heads, can be easily deployed using 8-way tensor parallelism, which is also the requirement for the Llama-70B model. In contrast, an OPT-125m model with 12 attention heads cannot be deployed across 8 GPUs, as discussed in § 5.1. We deploy the OPT-125m predictor solely on 1 GPU, necessitating the other 7 GPUs to wait when executing the predictor. This configuration results in a waste of resources and may lead to performance degradation.

Table 8: Relationship Between ListMLE Loss and Kendall's Tau

| Kendall's Tau  | 125m-OPT | 350m-OPT |
|----------------|----------|----------|
| ShareGPT       | 0.55     | 0.54     |
| LMSYS-Chat-1M  | 0.64     | 0.62     |

# G    Consideration of Ignoring The Prompt Length

In practice, we have found that focusing solely on the generated length is both simple and sufficiently effective.

First, our observations from the Imsys-chat-1M and ShareGPT traces, which represent real-world scenarios, indicate that prompt length is not a critical factor in generation time. Specifically, the prefill time constitutes only 5% on Imsys-chat-1M and 8% on ShareGPT, respectively, of the entire generation time, indicating that they have a minor impact on overall latency. Note that there are already long prompts in the workloads we tested. For example, 1% of all prompts in the ShareGPT dataset exceed 900 tokens.

Second, although this paper does not particularly focus on long contexts (e.g., prompt length > 32k tokens), we argue that handling long prompts is relatively straightforward. Since prompt lengths are always known a priori, it is easy to accurately approximate the latency of the prefill phase through profiling. We can also map the relative ranking of generation length into a length estimation based on the dataset distribution. By simply adding the prefill time estimation to the current framework, we can provide an end-to-end generation time approximation for scheduling.

# H Influence of Correcting Mispredictions Dynamically

We have implemented preemptive scheduling, where at each decoding step, we compare the generation rankings of new-incoming requests with those of the currently running requests and preempt those with lower rankings (as detailed in Algorithm 1). However, we do not re-predict the scores for requests that have already been executed during the generation process. Our findings, as presented in Tab. 9, indicate that re-prediction offers minimal improvement. These experiments were conducted using a Llama-3-8B model on a single 80GB A100 GPU.

Table 9: Influence of Correcting Mispredictions

| Latency (s/token) | Ours | Re-Prediction |
|---|---|---|
| ShareGPT | **0.43** | 0.44 |
| LMSYS-Chat-1M | **0.64** | 0.64 |

