# OpenReview forum: "Efficient LLM Scheduling by Learning to Rank"
_NeurIPS.cc/2024/Conference — NeurIPS 2024 poster_

### Official Review · Reviewer_JCjf · 2024-07-07

**Soundness:** 3
**Presentation:** 3
**Contribution:** 3
**Rating:** 6
**Confidence:** 4

**Summary:**

This paper proposes a learning-based rank predictor for scheduling LLM inference to reduce Head-of-Line (HoL) blocking issues, which significantly outperforms state-of-the-art LLM serving systems.

**Strengths:**

1. This paper addresses an important question in LLM serving.
2. This paper is easy to follow with a good presentation.
3. The evaluation results are comprehensive and solid.

**Weaknesses:**

1. One potential issue with preemptive scheduling for LLM inference is the accumulated unused KV cache. How do you handle them when the GPU reaches the maximum memory limit?

2. How much does the ranking model (OPT) size affect the prediction and throughput performance? For example, what if I use a smaller auxiliary model (OPT-125M) for a larger LLM (LLaMA-70B)?

3. How much is the performance gap between the ranking-based method and Oracle? It would be better if the authors could add such results to provide a performance upper bound.

**Questions:**

Please see the weaknesses above.

**Limitations:**

Please see the weaknesses above.

---

> ### Author Rebuttal · Authors · 2024-08-06
>
> We thank the reviewer for the constructive and detailed suggestions. We appreciate your generous comments!
>
> **Q1: One potential issue with preemptive scheduling for LLM inference is the accumulated unused KV cache. How do you handle them when the GPU reaches the maximum memory limit?**
>
> To address the issue of an increasing size of KV Cache, we will swap the requests to host memory (as vLLM [1] does) if the requests are not in the current step’s running batch (i.e., B in Algorithm 1) and cannot fit in the GPU memory. However, our experiments handling a burst of 2000 requests with a single llama-3-8b model on a 40GB A100 GPU and with a llama-3-70b model on 8 A100 GPUs resulted in almost no swapping using both FCFS and our scheduling, with vLLM’s default setting of a max batch size of 256 requests running simultaneously. This indicates that our method will not incur more preemptions than FCFS. This is because both our method and FCFS have a fixed execution order for all arriving requests at the beginning. However, our baseline MLFQ will adjust the execution order and call thousands of swaps with the same workload.
>
> **Q2: How much does the ranking model (OPT) size affect the prediction and throughput performance? For example, what if I use a smaller auxiliary model (OPT-125M) for a larger LLM (LLaMA-70B)?**
>
> Our results show that the model size has a minor effect on the prediction ability, as indicated in the following table:
>
> | Kendall’s Tau (↑) | 125m-OPT | 350m-OPT |
> |  ----  | ----  | ---- |
> | ShareGPT | 0.55 | 0.54 |
> | Lmsys | 0.64 | 0.62 |
>
> Using an OPT-350m is more for deployment considerations. The OPT-350m model with 16 attention heads is easy to deploy with 8-way tensor parallelism, as the llama-70b model requires 8-way tensor parallelism, while an opt-125m with 12 attention heads cannot be deployed on 8 GPUs, as discussed in Section 5.1. We deploy the opt-125m predictor only on 1 GPU, requiring the other 7 GPUs to wait when executing the predictor, which is a waste of resources and may lead to performance degradation.
>
> **Q3: How much is the performance gap between the ranking-based method and Oracle? It would be better if the authors could add such results to provide a performance upper bound.**
>
> The performance gap between the ranking-based method and the Oracle depends on the evaluation dataset. On some datasets, our proposed method can be as good as the Oracle. Due to noise and randomness in the sampling process, we define the Oracle as using sampling results with one seed to guide the schedule of another sampling, which is the best we can achieve knowing one sampling result.
> For example, tested on the Alpaca dataset and llama3-7b model, our proposed method can be very close to the Oracle regarding Kendall’s Tau and end-to-end latency of a burst of 2K requests. We tested the end-to-end performance on a single A100 80G GPU.
>
> | | Tau |  Latency (s/token)
> |  ----  | ----  | ----  |
> | Ours | 0.73 | 0.28 |
> | Oracle | 0.72 | 0.24 |
> | FCFS | / | 1.36 |
>
> On datasets like lmsys-chat-1m and ShareGPT, there is still a small gap between the proposed ranking-based method and the Oracle. We present in Table 3 the comparison between the ranking-based method (line Ranking (Ours)) and the Oracle (line Optimal Prediction).
>
> [1] Kwon, Woosuk, et al. "Efficient memory management for large language model serving with pagedattention." Proceedings of the 29th Symposium on Operating Systems Principles. 2023.

---

> > ### Comment · Reviewer_JCjf · 2024-08-07
> >
> > I would like to thank the authors for their detailed responses.
> >
> > My only suggestion would be to add Oracle as an additional baseline in the main evaluation if the paper is accepted.

---

> ### Author Response · Authors · 2024-08-08
>
> Dear Reviewer JCjf,
>
> We sincerely thank you for the prompt response. We greatly appreciate your constructive and insightful suggestions. We will keep Oracle in the main evaluation if the paper is accepted.
>
> Best Regards,
>
> Paper 5119 Authors

---

### Official Review · Reviewer_GSuM · 2024-07-08

**Soundness:** 3
**Presentation:** 3
**Contribution:** 2
**Rating:** 6
**Confidence:** 3

**Summary:**

This paper proposes an approach for optimizing scheduling in LLM  serving by learning a generated token length ranking model. The authors demonstrate that understanding the relative order of generation lengths can effectively guide the scheduling process, specifically through the use of SJF/ SRTF scheduling strategies.

**Strengths:**

1. The paper is well-written, making the methodology and results clear and easy to understand.
2. The experiments are well-designed and convincingly demonstrate the benefits of the proposed approach.
3. The proposed method has shown practical improvements when integrated with current serving techniques.

**Weaknesses:**

1. While the approach is effective, it builds upon existing work that has already identified the benefits of SJF/SRTF scheduling for LLMs[1][2]. The novelty is somewhat limited to the application of ranking loss instead of classification loss.
2. If we directly predict the token length, it could potentially offer advantages such as improved memory allocation and cache strategy adjustments, which are also crucial for optimizing LLM serving. In contrast, using relative order may not provide these benefits.
3. The paper lacks a thorough discussion of some related work,  such as [1][2]

[1] Efficient Interactive LLM Serving with Proxy Model-based Sequence Length Prediction

[2] Power-aware Deep Learning Model Serving with µ-Serve

**Questions:**

This paper only considers the generated length, which may affect the execution time for each query. However, prompt length also influences execution time. Wouldn't it be more reasonable to also take prompt length into consideration?

**Limitations:**

see weakness above.

---

> ### Author Rebuttal · Authors · 2024-08-06
>
> We thank the reviewer for the very insightful and helpful comments! We would like to address your questions in the below response.
>
> **Q1: While the approach is effective, it builds upon existing work that has already identified the benefits of SJF/SRTF scheduling for LLMs[1][2]. The novelty is somewhat limited to the application of ranking loss instead of classification loss.**
>
> We argue that our work presents the following novelties:
>
> First, recognizing the importance of relative length ranking instead of accurate token length after the introduction of paged memory allocation (proposed by [3]) is novel.
>
> Second, employing learning-to-rank to optimize the prediction is novel and outperforms previous methods.
>
> Third, the consideration of the fairness metric in LLM serving and starvation prevention is novel.
>
> **Q2: If we directly predict the token length, it could potentially offer advantages such as improved memory allocation and cache strategy adjustments, which are also crucial for optimizing LLM serving. In contrast, using relative order may not provide these benefits.**
>
> Predicting the number of output tokens may aid in more precise memory allocation, but we observe that in the presence of paged memory allocation (as proposed by vllm[3] and populated in all current LLM serving systems), the benefit diminishes, as paged attention reduces memory waste to as low as 5%.
> Despite memory allocation, knowing the exact number of output tokens in advance may reduce the likelihood of preemptions caused by kv-cache overrun. However, we found that this preemption cost can largely be mitigated (e.g., see FastServe[4]). Therefore, we pursue relative ranking prediction, which is a simpler task but suffices for approximating optimal scheduling.
>
> **Q3: The paper lacks a thorough discussion of some related work, such as [1][2]**
>
> We will include discussions with [1][2] in the related work section in a revised version. [1][2] are both concurrent works that use predictors to approximate SJF/SRTF. Both [1][2] propose a regression-based method for length prediction, fine-tuning a Bert model on the Lmsys-Chat-1M dataset with a regression L1 loss to predict the exact generation length. They tested models ranging from 300M to 3b and applied various batching policies (i.e., no batching, dynamic batching, continuous batching). The proposed method significantly improves latency and throughput under these settings. Additionally, it supports multi-round LLM conversations. Unlike this method, our proposed method is built on vLLM with paged attention and uses ranking loss to optimize the predictor model. We designed a preemptive scheduling method with starvation prevention to optimize the end-to-end performance of real-world LLM serving systems.
>
> **Q4: This paper only considers the generated length, which may affect the execution time for each query. However, prompt length also influences execution time. Wouldn't it be more reasonable to also take prompt length into consideration?**
>
> Considering prompt length is indeed a nice suggestion, which we have previously considered, but we found that in practice, focusing only on generated length is simple yet sufficiently effective.
>
> First, we observe from lmsys-chat-1m and ShareGPT – two traces that represent real-world scenarios – that prompt length is not crucial in generation time. Specifically, the prefill time accounts for only 5% on lmsys-chat-1M and 8% on ShareGPT, respectively, of the entire generation time, indicating they have a minor impact on overall latency. Note that there are already long prompts in the workloads we tested. For example, 1% of all prompts in the ShareGPT dataset exceed 900 tokens.
>
> Second, although this paper does not particularly focus on long context (e.g., prompt length > 32k tokens), we argue it is relatively straightforward to handle long prompts. Since prompt lengths are always known a priori – it is easy to accurately approximate the latency of the prefill phase via profiling. We can also map the relative ranking of generation length into a length estimation according to the dataset distribution. Then, simply adding prefill time estimation to the current framework together provides an end-to-end generation time approximation for scheduling.
>
>
> [1] Qiu, Haoran, et al. "Efficient interactive LLM serving with proxy model-based sequence length prediction." arXiv preprint arXiv:2404.08509 (2024).
>
> [2] Qiu, Haoran, et al. "Power-aware Deep Learning Model Serving with μ-Serve." 2024 USENIX Annual Technical Conference (USENIX ATC 24). 2024.
>
> [3] Kwon, Woosuk, et al. "Efficient memory management for large language model serving with pagedattention." Proceedings of the 29th Symposium on Operating Systems Principles. 2023.
>
> [4] Wu, Bingyang, et al. "Fast distributed inference serving for large language models." arXiv preprint arXiv:2305.05920 (2023).

---

> > ### Comment · Reviewer_GSuM · 2024-08-12
> >
> > Thanks for your response, my concern has been resolved, I will raise my score.

---

> > > ### Author Response · Authors · 2024-08-13
> > >
> > > Dear Reviewer GSuM,
> > >
> > > We sincerely thank you for raising your score. We highly appreciate your insightful and helpful comments. We will incorporate the new discussions into our final revised manuscript.
> > >
> > > Best Regards,
> > >
> > > Paper 5119 Authors

---

### Official Review · Reviewer_GoTD · 2024-07-08

**Soundness:** 3
**Presentation:** 3
**Contribution:** 3
**Rating:** 6
**Confidence:** 3

**Summary:**

This paper reveals the Head-of-Line (HOL) blocking problems caused by the first-come-first-serve (FCFS) scheduling strategy in LLM services. To alleviate these problems, the authors train an OPT model to generate scores for evaluating the relative text length of given prompts. Based on these scores, the authors develop a novel scheduler for LLM inference and serving. Experimental results demonstrate the effectiveness of the proposed method, significantly outperforming the baseline method.

**Strengths:**

1. The proposed method is efficient and effective. Training a small language model (i.e., a 125M OPT model) is cheap, and the resulting latency gains are substantial.
2. This paper is novel. Unlike traditional methods that predict the real generation length, predicting the relative ordering between request lengths is sufficient for ranking.

**Weaknesses:**

1. Since the request queue Q is re-ranked after each batch of data is scored, the ranking scheduler may be sensitive to the batch size.

**Questions:**

1. Could you give a more detailed analysis of the relationship between ListMLE loss and Kendall’s Tau coefficient?
2. Are all the last hidden states of the OPT model used to map to a score, or are only specific hidden states of a token used? Using a decoder-only model to extract features of a text seems unusual.

**Limitations:**

See weaknesses and Questions.

---

> ### Author Rebuttal · Authors · 2024-08-06
>
> We thank the reviewer for the insightful and helpful feedback! We address all your questions and concerns below.
>
> **Q1: Since the request queue Q is re-ranked after each batch of data is scored, the ranking scheduler may be sensitive to the batch size.**
>
> **A1:** The ranking scheduler is not sensitive to the batch size. We compare the sensitivity of the scheduler to the batch size as follows. We use the predictor to compute tau with different batch sizes and obtain the mean and variance across the entire test set.
>
> | Batch size  | Kendall's Tau mean | Kendall's Tau variance |
> |  ----  | ----  | ----  |
> | 8 | 0.619 | 0.04 |
> | 16 | 0.625 | 0.02 |
> | 32 | 0.624 | 0.008 |
> | 64 | 0.625 | 0.0007 |
> | 128 | 0.619 | 0.001 |
>
> With different batch sizes, Kendall’s Tau maintains a small range. Additionally, our method addresses severe head-of-line (HOL) problems when there are numerous requests, in which case, the batch size is often sufficiently large for the predictor to be effective and robust.
>
> **Q2: Could you give a more detailed analysis of the relationship between ListMLE loss and Kendall’s Tau coefficient?**
>
> **A2:** ListMLE loss and Kendall’s Tau coefficient are highly negatively correlated. We demonstrate this relationship by recording Tau and loss during the training process. The Pearson correlation coefficient is -0.9:
>
> | Step  | Tau  | Loss  |
> |  ----  | ----  | ----  |
> | 20 | 0.44 | 77.79 |
> | 40 | 0.51 | 75.73 |
> | 60 | 0.53 | 72.61 |
> | 80 | 0.54 | 70.14 |
> | 100 | 0.55 | 70.59 |
> | 120 | 0.53 | 70.09 |
> | 140 | 0.56 | 67.01 |
> | 160 | 0.59 | 69.94 |
> | 180 | 0.59 | 70.88 |
> | 200 | 0.57 | 68.84 |
> | 220 | 0.59 | 68.67 |
> | 240 | 0.61 | 66.90 |
> | 260 | 0.58 | 67.23 |
> | 280 | 0.56 | 68.71 |
>
>
> **Q3: Are all the last hidden states of the OPT model used to map to a score, or are only specific hidden states of a token used? Using a decoder-only model to extract features of a text seems unusual.**
>
> **A3:** We use the last token’s hidden states of the OPT model for two reasons.
>
> First, the OPT model provides similar performance compared to the encoder model DistilBert, as shown in the following evaluation:
>
> | Kendall’s Tau (↑) | OPT | DistilBert |
> |  ----  | ----  | ----  |
> | ShareGPT | 0.54 | 0.52 |
> | Lmsys | 0.62 | 0.62 |
>
> Second, the hidden states of the last token of a decoder model have been previously shown to be predictive of text features. For example, Representation Engineering [1] demonstrates that we can extract attributes such as honesty, emotions, fairness, and more from the hidden states of the last token of the decoder model.
>
>
>
> [1] Zou, Andy, et al. "Representation engineering: A top-down approach to ai transparency." arXiv preprint arXiv:2310.01405 (2023).

---

> > ### Comment · Reviewer_GoTD · 2024-08-13
> >
> > Thanks for your detailed explanation and additional experiments, which address my majority concerns.
> > Furthermore, I believe it would be great if provided a theoretical analysis of the relationship between the ListMLE loss and Kendall’s Tau.

---

> > > ### Author Response · Authors · 2024-08-14
> > >
> > > Dear Reviewer GoTD,
> > >
> > > Thank you for your constructive and insightful suggestions. The ListMLE loss defines a parameterized exponential probability distribution over all scores (as given by the model) and formulates the loss function as the negative log likelihood of the ground truth ranking $y$. Meanwhile, Kendall's Tau measures the ordinal association between the scores (as given by the model) and the ground truth ranking $y$. It is challenging to accurately describe the relationship between the likelihood and ordinal association. However, we provide an analysis demonstrating that minimizing the ListMLE loss can help improve Kendall’s Tau.
> > >
> > > To simplify the problem, we assume there are no ties between any two items, meaning each pair should be either concordant or discordant. In this case, Kendall's Tau is defined as $\tau=\frac{N_c-N_d}{n(n-1)/2}$, where $N_c$ and $N_d$ are the number of concordant and discordant pairs in two rankings, and $n$ is the total number of items. As $N_d$ increases, $N_c$ decreases because the sum of $N_c$ and $N_d$ is fixed. Consequently, we have $\Delta \tau=\frac{4\Delta N_c}{n(n-1)}$, where $\tau$ increases when $N_c$ increases.
> > >
> > > ListMLE loss is defined as $\mathcal{\phi}(g(x),y)=-\log P\left(y \mid x ; g\right)$, where $P(y \mid x ; g)$ represents the likelihood of the ground truth ranking $y$. As the likelihood of the ground truth ranking $y$ increases, the loss decreases. Although the increase of $P(y \mid x ; g)$ does not guarantee that $N_c$ increases, the increase in the likelihood of the ground truth ranking should generally lead to a greater agreement between the ground truth ranking and the scores given by the model, which implies an increase in the number of concordant pairs (or $N_c$) and a decrease in the number of discordant pairs (or $N_d$) between the scores and the ground truth. Thus, minimizing the loss can help improve Kendall’s Tau.
> > >
> > > We hope our response addresses your concerns.
> > >
> > > Best Regards,
> > >
> > > Paper 5119 Authors

---

> > > > ### Comment · Reviewer_GoTD · 2024-08-14
> > > >
> > > > Thanks for your additional explanation. I would like to slightly raise my score to 6.

---

> > > > > ### Author Response · Authors · 2024-08-14
> > > > >
> > > > > Dear Reviewer GoTD,
> > > > >
> > > > > Thank you for increasing your score. We greatly appreciate your insightful and constructive comments. We will incorporate the new discussions and results into our final revised manuscript.
> > > > >
> > > > > Best Regards,
> > > > >
> > > > > Paper 5119 Authors

---

### Official Review · Reviewer_RLWW · 2024-07-13

**Soundness:** 2
**Presentation:** 3
**Contribution:** 2
**Rating:** 4
**Confidence:** 4

**Summary:**

The paper addresses the inefficiencies in scheduling LLM inference requests, which often use a first-come-first-serve (FCFS) strategy, leading to Head-Of-Line (HOL) blocking and reduced throughput. The authors propose a novel scheduling method based on predicting the relative ranks of output lengths in a batch of requests, rather than attempting to predict exact generation lengths. This prediction helps in approximating the shortest-job-first (SJF) schedule, which is known to minimize average latency.

**Strengths:**

The paper employs a straightforward but effective scheduling algorithm that approximates the shortest job first (SJF) strategies. This approach effectively reduces response latency and improves throughput. The authors have tackled the challenge of accurately approximating SJF. The empirical results demonstrate significant improvements in both latency and throughput, highlighting the effectiveness of their approach. The paper introduces interesting metrics to determine the relative range of output lengths.

The paper addresses a crucial issue in LLM workload scheduling. By focusing on reducing response latency and enhancing throughput, it tackles a significant problem that is highly relevant to the efficiency and performance of LLM servingsystems.

**Weaknesses:**

- The current scheduling approach only considers output length. Would you also consider other dimensions, such as prompt length? Longer prompt lengths can consume more memory and increase token latency, impacting overall response latency and throughput. Additionally, would you consider implementing preemptive scheduling to correct any mispredictions dynamically?

- Your predictor is trained using 10k traces from ShareGPT and LM-SYS. However, these traces are primarily from GPT-4 and other models. Have you considered that different models Llama3 might behave differently, with varying verbosity and output lengths even for the same prompts? If the predictor cannot be reused across different models, you might need to account for the overhead of retraining the model to maintain accuracy.

- You should discuss Andes [1], which also propose a request scheduling strategy to improve quality of experience.
[1] Andes: Defining and Enhancing Quality-of-Experience in LLM-Based Text Streaming Services

- SJF scheduling inherently risks starving requests with longer response length, as these jobs can be indefinitely delayed. How do you address this issue to ensure that longer requests are also processed in a timely manner?

**Questions:**

1. Why is the improvement on dataset Sharegpt and lmsys different, as shown in table3.

**Limitations:**

See weakness.

---

> ### Author Rebuttal · Authors · 2024-08-06
>
> We thank the reviewer for the insightful and helpful feedback! We would like to address your questions in the below response.
>
> **Q1.1: The current scheduling approach only considers output length. Would you also consider other dimensions, such as prompt length? Longer prompt lengths can consume more memory and increase token latency, impacting overall response latency and throughput.**
>
> **A1.1:** Considering prompt length is indeed a nice suggestion, which we have previously considered, but we found that in practice, focusing only on generated length is simple yet sufficiently effective.
>
> First, we observe from lmsys-chat-1m and ShareGPT – two traces that represent real-world scenarios – that prompt length is not crucial in generation time. Specifically, the prefill time accounts for only 5% on lmsys-chat-1M and 8% on ShareGPT, respectively, of the entire generation time, indicating they have a minor impact on overall latency. Note that there are already long prompts in the workloads we tested. For example, 1% of all prompts in the ShareGPT dataset exceed 900 tokens.
>
> Second, although this paper does not particularly focus on long context (e.g., prompt length > 32k tokens), we argue it is relatively straightforward to handle long prompts. Since prompt lengths are always known a priori – it is easy to accurately approximate the latency of the prefill phase via profiling. We can also map the relative ranking of generation length into a length estimation according to the dataset distribution. Then, simply adding prefill time estimation to the current framework together provides an end-to-end generation time approximation for scheduling.
>
> **Q1.2: Additionally, would you consider implementing preemptive scheduling to correct any mispredictions dynamically?**
>
> **A1.2:** We’ve already employed preemptive scheduling. In each decoding step, we compare the generation rankings of new-coming requests and the running requests and preempt the requests with low rankings (Algorithm 1).  However, we will not re-predict the score for an executed request during the generation process. We found that re-prediction yields little improvement, as shown in the following table. We conducted the experiments using a llama-3-7b model with a single 80GB A100 GPU.
>
> |  Latency(s/token)   | ours  | re-predict |
> |  ----  | ----  | ----  |
> | lmsys  | 0.43 | 0.44 |
> | ShareGPT  | 0.64 | 0.64 |
>
> **Q2: Your predictor is trained using 10k traces from ShareGPT and LM-SYS. However, these traces are primarily from GPT-4 and other models. Have you considered that different models Llama3 might behave differently, with varying verbosity and output lengths even for the same prompts? If the predictor cannot be reused across different models, you might need to account for the overhead of retraining the model to maintain accuracy.**
>
> **A2:**  We would like to clarify that we do not use the outputs from GPT-4 or other models. The prompts are collected from human users, and answers are generated by the target model (e.g.,  llama3), as stated in section 4.2. We only need to train predictors for each dataset-model pair. The cost of training is negligible: It takes ~10 minutes on a single A100 GPU to train a predictor per long-standing serving job.
>
> **Q3: You should discuss Andes [1], which also propose a request scheduling strategy to improve quality of experience. [1] Andes: Defining and Enhancing Quality-of-Experience in LLM-Based Text Streaming Services**
>
> **A3:** Andes and our proposed method focus on different aspects of LLM serving. Andes introduces a novel quality of experience (QoE) metric for text streaming services, which measures human satisfaction during the entire end-to-end token delivery process. Andes employs an online preemptive scheduling method. At the beginning of each time quantum, it decides which requests to run based on the scheduling objectives (e.g., average QoE) for the upcoming time frame. However, our proposed method primarily aims to optimize latency by executing the requests with the lowest estimated generation length (as predicted by the predictor) to approximate SJF/SRTF at the start of each decoding step. We will discuss Andes in the related work section in the revised version.
>
> **Q4: SJF scheduling inherently risks starving requests with longer response length, as these jobs can be indefinitely delayed. How do you address this issue to ensure that longer requests are also processed in a timely manner?**
>
>
> **A4:** We have discussed starvation prevention in section 4.3 and presented the results in section 5.5. Specifically, we defined max_waiting_time fairness to measure whether a request suffers from starvation. Our starvation counter (in Algorithm 1) prevents users from waiting too long for responses. The results in section 5.5 (Figures 4 and 5) demonstrate that our method can significantly alleviate starvation.
>
> **Q5: Why is the improvement on dataset Sharegpt and lmsys different, as shown in table3.**
>
> **A5:** Different datasets exhibit varying prompt/generation length distributions (Appendix B) and data distributions. For instance, some problems can be answered fixedly, resulting in LLMs producing similar generation lengths across different trials. However, other problems may be addressed in various ways, leading to LLMs generating very different lengths in different trials. The latter pattern is unpredictable and ShareGPT encompasses more of these unpredictable problem types. Line *Optimal Prediction* in Table 3 employs the generation length of one random seed to predict the length using another seed. It shows that predicting the generation length of the ShareGPT dataset is more challenging. Consequently, our predictor performs less effectively on the ShareGPT dataset.

---

### Decision · Program_Chairs · 2024-09-25

**Decision:**

Accept (poster)

**Comment:**

This paper discusses the problem of scheduling different jobs for LLMs, describing a policy to make it more efficient.
Reviewers found the proposed method effective, the problem important, and the paper clear. Concerns regarding overhead, novelty, and sensitivity to the batch size were largely addressed during the rebuttal.